# How do gender disparities in entrepreneurial aspirations emerge in Pakistan? An approach to mediation and multi-group analysis

**Ghulam Raza Sargani** [1]*, **Yuansheng Jiang**[1]*, **Deyi Zhou**[2], **Abbas Ali Chandio**[1], **Mudassir Hussain**[3], **Asif Ali**[4], **Muhammad Rizwan**[5], **Najeeb Ahmed Kaleri**[6]

1 College of Economics, Sichuan Agricultural University, Sichuan, Chengdu, PR China, 2 School of Economics & Management, Huazhong Agricultural University, Hubei, Wuhan, PR China, 3 Department of Education and Research, University of Lakki Marwat KPK, Pakistan, 4 Institute of Business Administration, University of Sindh, Jamshoro, Sindh, Pakistan, 5 School of Economics & Management, Yangtze University, Hubei, PR China, 6 Sindh Horticulture Research Center, Mirpurkhas, Sindh Pakistan

* razasargani@sicau.edu.cn (GRS); yjiang@sicau.edu.cn (YJ)

**Data Availability Statement:** All relevant data are within the manuscript and its Supporting Information files.

## Abstract

This cross-sectional study sought to identify gender differences in individual behavioral attitudes, personal traits, and entrepreneurial education based on planned behavior theory. The Smart partial least squares (PLS) structural equation model and PLS path modeling were used. A survey design was used to collect data from 309 samples using quantitative measures. The model was tested for validity and reliability and showed variance (full, $R^2 =$ 58.9% and split, $R^2 = 62.7\%$ and R2 = 52.7%) in male and female model predictive power, respectively. Subjective norms (SN), personality traits (PT), and entrepreneurial education (EE) significantly impacted the male sample's intention. Females' intentions toward entrepreneurship was less affected by attitude toward behavior (ATB), subjective norms (SN), perceived behavioral control (PBC), and entrepreneurship education (EE). Further, attitudes, social norms, and behavioral controls as mediation variables indicate a significant and positive role of male and female intentions. These findings imply that behavioral beliefs (ATB, PBC, and SN) influence entrepreneurial intention-action translation. The results significantly supported the designed hypotheses and shed light on individual personality traits (PT) and entrepreneurship education (EE) underpinning enterprise intention. The study determined that EE and PT are the strongest predictors of intention, thus highlighting the role of these motives in the entrepreneurial process. This study contributes to the growing body of knowledge on youth entrepreneurs, expands our understanding of entrepreneurship as a practical career choice, and offers a novel account differentiating male and female PT. The drive to evaluate the effects of entrepreneurial intention among budding disparities in Pakistan requires a more profound knowledge of the aspects that endorse entrepreneurship as a choice of profession and enhances youth incentive abilities to engage in entrepreneurial activities based on exploitation.

**Funding:** The authors received no specific funding for this work.

**Competing interests:** The authors have declared that no competing interests exist.

## Introduction

Entrepreneurship is critical for economic development and well-being. Entrepreneurs' participation in a country's economic growth is crucial because of their essential role in development [1]. Establishing enterprises has been sluggish in the Pakistani context and is essential for building a robust entrepreneurial environment for nascent entrepreneurs [2]. Usually, women face significant opposition when establishing their own business, which leads to the perception that considered as food provider instead of food creators [3]. As women encounter increasing obstacle to careers that pay, motivation and encouragement may enhance their performance in business intentions because they represent unexploited assets that can be used to improve a country's economic prospects [4]. Some studies found that obtaining and using the enterprise economy relies on women's entrepreneurship growth [2, 5]. As a result, start-up activity has increased owing to new ventures that meet unmet customer needs across the nation [6]. Nevertheless, universities and academia provide entrepreneurship opportunities, particularly fewer in the agricultural sector, since they make it possible for consumers to test and incubate new businesses and acquire entrepreneurial skills. The impact of the entrepreneurial class is seen in the mindsets of university students who see themselves as business owners [3].

Pakistan is an agrarian country and home to approximately 200 million people, with 6.14% youth unemployment [7]. However, poverty and unemployment are chronic problems and have become a great concern in Pakistan [3]. Agricultural students, in particular, endure challenging circumstances and face social, economic, environmental, and political non-compensation hitches [2, 7]. To promote an entrepreneurship environment for job creation and poverty alleviation problems in underdeveloped nations [3]. Prior research has shown that thousands of university graduates have joined the labor market every year. However, their employment requirements are not met by the market capacity [3, 8]. The increasing unemployment levels of educated individuals, particularly farm businesses, have become a great concern today [3]. It is a cause of fear for managers to deal with new jobs, particularly in Sindh/Pakistan [3, 9].

Despite the rapidly growing interest in entrepreneurial research in recent years, there is still a dearth of knowledge regarding entrepreneurship in transitioning emerging countries [10, 11] There are significant disparities between developing and established economies regarding institutional infrastructure, entrepreneurial education methods, and stringent regulatory frameworks [12]. The collectivistic culture of the nation distinguishes the setting from the majority of current research, which has been performed in western countries, as previously stated. entrepreneurship education (EE) in Pakistan is still in its infancy and has received little attention. Failure to recognize this diversity has significant implications for entrepreneurial policy and human development, especially about gender. The research is being performed in Pakistan, an entrepreneurial environment is favorable to job creation and poverty reduction in emerging nations such as Pakistan [3]. Despite increasing entrepreneurship research and a transition from ancient to modern businesses, there is still a dearth of evidence on how intentions and values emerge throughout entrepreneurial forms. However, Pakistan's entrepreneurial environment remains unexplored. The nation continues to fall below peers in terms of young people's proclivity for and retention of entrepreneurial activity, especially in the agriculture sector, which has not been investigated yet.

Thus, this paper treats the complexities of EE. Personality traits (PT) should be based on entrepreneurship theory and implemented in student-focused learning activities, setting the stage for a focus on student-oriented learning disparities if the students' construction of entrepreneurship differs from the frames in Pakistan. The present research applies the theory of planned behavior's (TPB) conceptual and intentional models to ascertain gender differences in entrepreneurship intention (EI) emerging in Pakistan. This cross-sectional study aimed to

discover gender variations in individual PT, EE), and TPB antecedents. That is, attitude toward behavior (ATB), subjective norms (SN), and perceived behavioral control (PBC), serve as mediator indicators among male and female aspirants. The Smart PLS structural equation model (SEM) and partial least square path modeling were used.

## Theory-based framework and hypotheses development

The theory of planned behavior was first created to represent the connection between an individual's attitude toward a specific action and the actual behavior [13]. According to the TBP, intentions serve as a link between attitudes and behaviors. TPB reflects variables such as the desire for entrepreneurship (personal attitude), the social acceptability of entrepreneurship to a normative reference group (subjective norms), and the perceived feasibility and control of becoming an entrepreneur (perceived behavioral control). Understanding individual decision-making is critical in fostering an entrepreneurial spirit; the TPB recognizes an insatiable intention to engage in an activity as an instant antecedent [13]. As summarized by the TPB, individuals will participate in a behavior if they want to do so, have the opportunity and resources, assess it favorably, believe others think they should engage in it, and feel it is within their control [13].

### Entrepreneurial intention

Entrepreneurial intention (EI) is crucial in the entrepreneurial process because it lays the groundwork for effective entrepreneurial activities [14]. However, establishing a new business is a lengthy and complicated process that involves many mediating factors [15]. Several significant theoretical breakthroughs have been made in understanding the new entrepreneurial process [16]. Whereas the study of [17] found that intentions and behaviors influence real behavior. However although an individual's willingness to participate in entrepreneurial activity is influenced by three distinct attitudinal domains, each of which is focused on action [18] and according to the concept of planned behavior, one must first believe in oneself to change behavior [13, 19, 20],examine their willingness to take risky actions when establishing a new business or entrepreneurial activity by determining whether they express a belief that the behaviors are accurate and actionable.

Thus, entrepreneurship is a course of action that is influenced by an individual's personality. TPB shows that the more the individuals are willing to do the activity in question, the more favorable the attitude and subjective norm, and the greater the perceived behavioral control when people are given a sufficient degree of real influence over the behaviors of legitimate entrepreneurial careers. They are expected to pursue them [21] and [22] shows that entrepreneurial intentions are positively influenced by personal attitude and perceived behavioral control but not by subjective norms. The relationship between personal attitudes and social norms and intentions, but not between perceived behavioral control and choices validated by [23].

### Attitude toward behavior

Attitude and behavior denote the extent to which a person has a positive or disadvantageous assessment or appreciation of the action [13]. Personal desirability is the value placed on an entrepreneur's potential for significant professional achievement. In the context of entrepreneurship, attitude toward self-employment has been described as "the difference in views of personal attractiveness in being self-starting versus organizationally employed [24]. According to [19], an attitude toward start-up is the degree to which an individual has a favorable or negative personal value of becoming an entrepreneur; according to a prior study, ATB is the leading factor, including the aspirations of an entrepreneurial profession [15, 25]. Consequently, the strong connection between ATB and EIs is agreed upon in the context of the TPB [26–28] found that male fledgling entrepreneurs regarded financial success and innovation as

more important professional motivations for entrepreneurship than non-entrepreneurs. According to [29], women put a greater premium on non-wage elements of self-employment than men do, and [30] found that women are more receptive to entrepreneurship as a replacement for part-time labor. According to [31], women become entrepreneurs to balance job and family obligations, while men pursue wealth creation or economic growth.

## Subjective norm

Subjective norms reflect normative views regarding entrepreneurship as a career option weighted by a desire to conform to these normative beliefs [23] discovered a substantial but insignificant effect of subjective norms on entrepreneurial aspirations. [32] suggested that this abysmal connection may be ascribed to a few people whose behavior is mainly motivated by perceived social pressure. [33] argue that in certain circumstances, females may be more influenced by societal forces than males. Positive role models, according to [34], have a greater influence on female career choices in a male-dominated discipline, such as engineering. Subjective norms are often shown to be poor predictors of TPB models [35]. The research showed that SN is important in altering EIs and was insignificant [22, 36, 37]. According to prior research, we anticipate that women will be more receptive to conforming to normative referents than their male counterparts. Researchers suppose that men and women have the same normative view of important people, but women will be more driven to comply with these referents.

## Perceived behavioral control

Perceived behavioral control is a function of the significance weighted by the power of control beliefs toward establishing an enterprise. The PBC notion, originally presented by Ajzen [13] as another antecedent component that may predict intention, was described as a person's impression of the ease or difficulty of doing the action of interest. Initially, [38] conceptualized PBC as a one-dimensional structure almost comparable to a social self-effectual learning structure and evaluation of an individual's ability to act necessary for the future. Numerous researchers have replaced PBC with self-efficacy with the idea that PBC and independence are fundamentally comparable building blocks [23, 39, 40]. This implies personal conviction in designing, implementing, and managing people's conduct [13]. Individuals need to explain whether the entrepreneurial activity is comfortable or not [31, 41]. There were significant gender variations in the behavioral control emotions. According to [42] women's tendency to establish new companies is linked to awareness of current possibilities and self-assessment of ability and expertise. [41] discovered that women scored lower on entrepreneurial aspirations and internal control measures problem solving, decision making, money management, creativity, consensus building, and leadership. While external control beliefs, an individual might perceive financial resources as necessary to start a business [43]. We predict that prospective female entrepreneurs would place a premium on internal control feelings such as knowledge, the capacity to identify possibilities, and creativity when assessing the feasibility of becoming an entrepreneur.

We define the following hypothesis in line with the TPB framework:

**H1:** TPB components (a) ATB, (b) SN, and (c) PBC will have a significant positive impact on entrepreneurial intention (EI)

## Application of EE to TPB

EE helps customers develop entrepreneurial abilities, talents, and a broad range of professional skills [44]. Previous research has demonstrated that EI effectively inspires people to undertake an entrepreneurial career on purpose, convert into entrepreneurial behavior, and achieve better

entrepreneurship. [33] indicated that they had become entrepreneurs within 10 years of graduating from an entrepreneurship course. However, TPB provides such a proxy to evaluate and minimize the problems [40] proposed using the measured probability of company start-up as a proxy for educational benefits. It reflects a person's high probability of starting a company as an actual action to adhere to their goal [45]. Thus, by the TPB's three domains, entrepreneurial education determines an individual's entrepreneurial intention. As a result, the following hypotheses were developed:

**H2:** EE has a significant positive impact on (a) ATB, (b) SN, (c) PBC, and (d) EI.

## Application of personality traits to TPB

The TPB is an appropriate theoretical measurement instrument to investigate the effects of many factors, such as individual skills and personality characteristics, on EI precedents [46–48]. Personality traits (PT) are significant variances and significantly impact individuals who become self-employed [49, 50], distinguishing between entrepreneurs, and non-entrepreneurs [50, 51] with regard to the effects of PT on EI. The study of [40, 42] discussed risk propensity, ambiguity, and self-effectiveness tolerance [50, 52]. However, [14, 53] also assessed entrepreneurial passion, inventiveness, passion, and proactive personalities. Some studies [27, 54] verified the characteristics described as a particular characteristic of professional choices, called entrepreneurial qualities. These traits significantly impact corporate culture and EI [55], as a precedent for TPBs, indirectly explaining EIs. The direct and indirect relationship between individual traits and TPB dimensions continues to be unilateral [6, 56]. Following the processes outlined by career choice and the concept of internal and social adaptation, we expect that individuals will be attracted to entrepreneurship through an intimate connection between their personalities and the requirements of an enterprise. [6]. Therefore, the following hypothesis is developed:

**H3:** PT will have a significant positive effect on (a) ATB, (b) SN, (c) PBC, and (d) EIs.

## Mediation effects of the theory of planned behavior

The TPB antecedents have been employed as intermediates in numerous studies investigating cultural, psychological, and socio-economic factors [57, 58]. There is also an indirect relationship between personality and characteristics of entrepreneurial conduct mediated via numerous aspects, including attitudes and intentions [39, 59]. The business process is challenging to understand based on personality characteristics. A necessary explanation is given by understanding this process and the enabling function of the TPB and EE. Business intentions [16]. Previous research has shown how TPBs can predict characteristics, enterprise training, and entrepreneurship [10, 20]. Independent research on PT, EE, and TPB in the exercise area has been conducted. Consequently, it is unknown whether the relationship between personality and exercise behavior is mediated by social-cognitive conceptions, such as the TPB postulates depicted in Fig 1. Thus, the propositions may be formulated as

**H4:** There will be a significant positive impact of (a) EE, (b) PT, on EIs mediated by ATB.

**H5:** There is a significant positive impact of (a) EE, (b) PT, on EIs mediated by the SN.

**H6:** There will be a significant positive impact of (a) EE, (b) PT, on EIs mediated by PBC.

## Gender gaps in entrepreneurial environments

According to gender perspectives on entrepreneurship, women are less likely than men to prefer traditionally male-dominated occupations, owing to women's tendency to have lower self-

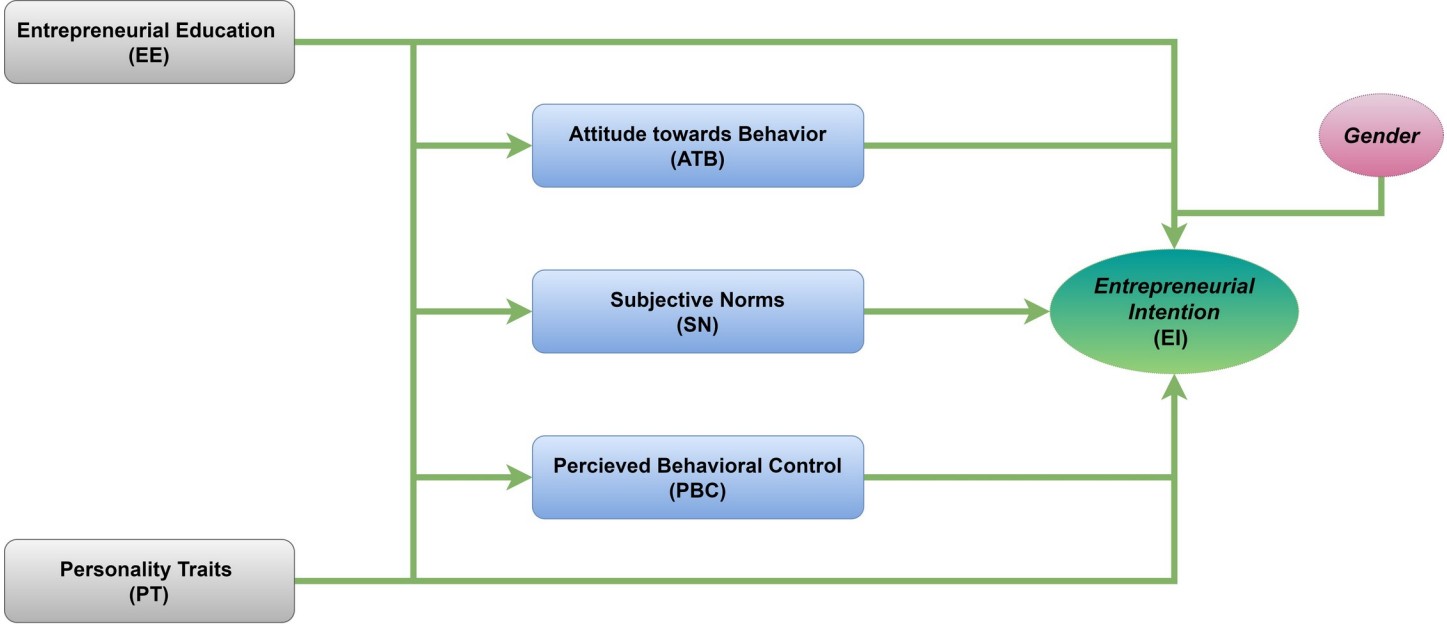

**Fig 1. Study analytic framework extended theory of planned behavior.**

efficacy perceptions regarding entrepreneurial career intentions and beliefs about social gender differences in personality and behavior. However, there are persistent gender disparities in entrepreneurship and self-employment worldwide [60]. Even though this is a significant problem, women's lower entrepreneurial activity inclinations are not well understood [61]. According to [61, 62], men are considered to be more entrepreneurially inclined. Nevertheless, [63] findings do not offer evidence but examine gender stereotypes, gender roles, and discrimination in market access are explained by various contextual variables [64]. According to [3, 27], women have a lower entrepreneurial inclination and trust in company skills, social network features, and a greater fear of failure. However, [65] believes that there is a link the intangible character of female entrepreneurs' credibility; it must be taken more seriously. However, the studies of [6, 66] revealed that women entrepreneurs in Pakistan do not have the same chances as men because of deeply ingrained discriminatory societal and cultural norms in support systems that assist these budding businesspeople. Women are most equipped to be homemakers in a male-dominated culture, which presents tremendous difficulties [6, 66]. Thus, male households seldom support female households, resulting in restricted regional agility, a lack of social capital, and gender disparities in entrepreneurship and venture potential among men and women [61, 62, 67]. Given the proposed model for the test in both gender groups, the hypothesis is as follows:

**H7.** In Pakistan, the effects of EE and PT are attributed to differences between men and women in the extent of the TPBs a) {ATB, SN, and PBC} on EIs.

## Material and methods

### Participant consent and ethical consideration

Before data collection, ethical approval and consent were obtained from the College of Economics Research Ethics Committee (Sichuan Agricultural University, Chengdu Campus,

China). Additionally, all the eligible respondents of both the survey and interview participants were informed about the study's aims, voluntary participation, and the right to withdraw at any time without giving a reason. They were assured that the information to be collected would be kept confidential and that the research would purely be used for academic purposes. The consent statement was included in writing on dummy-coded variables and the Likert Scale for the survey and for the interview protocols. During qualitative data collection, the researcher verbally shared the aims and objectives of the study with the participants in English.

## Target population

We conducted the study on 400 graduate students from Pakistan's four most prominent universities. Our selection of a simple random sampling strategy based on population-restricted agribusiness students' context's distinctive gender inequalities in entrepreneurship. We selected students interested in business who were about to make a career decision to ensure that the entrepreneurial aspirations construct included sufficient variance [22] to support the validity of the questionnaire based on the measurement techniques already used and verified [68]. In turn, the author performed pre-survey testing on 30 respondents to confirm that the questions were intelligible and reliable in this study. Some issues have been relocated and modified to simplify the research.

## Data collection procedure and participants

A total of 400 questionnaires were sent out, and 350 were returned, resulting in 309 valid questionnaires and a response rate of 77.25%; the study comprised participants from various agricultural sectors, with 184 males (60%) and 125 females (40%) participated in this study. When asked about the entrepreneurial intention EI, N = 65.4% responded "yes" and N = 34.6% replied with "no" intention in business start-ups. Although 62.5% said "yes" regarding the PEK factor, and 37.5% had no prior entrepreneurial knowledge, when enquired about prior entrepreneurial exposure PEX, approximately 57% answered "yes", and about 43% argued that they have no such exposure regarding business. Similarly, when asked about prior farming exposure PFE and prior entrepreneurial exposure PEE, approximately 64% had farm exposure, and 29.4% only show no farming experience in the line with almost 58% said that they have parental entrepreneurial exposure whereas almost 37% revealed that do not have any parental entrepreneurial exposure (see Table 1).

## Measures of sample constructs

The questionnaire used in this research was rated on a 5-point Likert scale and was adapted from prior literature related to the TPB. The degree of agreement ranged from 1 (strongly disagree) to 5 (strongly agree). Constructs of EI and PBC were assessed using five items derived from [19]. They used five questions to determine attitudes toward behavior (ATB), including four items of the SN scale, which was used for measurements drawn from [23]. Finally, EE was evaluated using five items, and PTs were measured using five questions adapted from [48].

## Control variables

Entrepreneurial intention (EI) has been linked to various control factors [19]. Dummy-coded variables (0 = female; 1 = male) and (1 = yes; 0 = no) were used for EI, prior entrepreneurial knowledge (PEK), parental entrepreneurial exposure (PEX), prior entrepreneurial exposure (PEE), and prior farming exposure entrepreneurial intention (EI), prior entrepreneurial knowledge (PEK), parental entrepreneurial exposure (PEX), and prior entrepreneurial

**Table 1. Demographic characteristics of the respondents.**

| Variable distribution | | (N) | (% age) | (M) | (STDEV) | Total |
|---|---|---|---|---|---|---|
| **Gender** | Male | 184 | 59.5 | 0.60 | 0.492 | 309 |
| | Female | 125 | 39.5 | | | |
| **Entrepreneurial Intention (EI)** | Yes | 202 | 65.4 | 0.65 | 0.477 | 309 |
| | No | 107 | 34.6 | | | |
| **Prior Entrepreneurial Knowledge (PEK)** | Yes | 193 | 62.5 | 0.62 | 0.485 | 309 |
| | No | 116 | 37.5 | | | |
| **Parental Entrepreneurial Exposure (PEX)** | Yes | 175 | 56.6 | 0.57 | 0.496 | 309 |
| | No | 134 | 43.4 | | | |
| **Prior Entrepreneurial Exposure (PEE)** | Yes | 179 | 57.9 | 0.58 | 0.494 | 309 |
| | No | 113 | 36.6 | | | |
| **Prior Farming Exposure (PFE)** | Yes | 196 | 63.4 | 0.63 | 0.482 | 309 |
| | No | 75 | 29.4 | | | |

Note: (N) = Number of samples, (% age) = Percentage (M) = Means, (STDEV) = Standard Deviation.

exposure (PFE) to determine gender disparities in entrepreneurial action and inclinations which also have been confirmed by the study of [69].

## Data analysis

The authors used statistical package for social sciences (SPSS- 25) and Smart PLS-3 to analyze data. The structural equation model (SEM) is an accurate method used to evaluate social, behavioral, and agricultural sciences to validate the connection of model variables [70] empirically. The adoption of this model comprises many continuous and preparatory phases, including a description of the theoretical model to be tested, parameter estimates, and assessments. The reflective measurement model was evaluated using two sets of criteria. The first set of item criteria was principal component factor loading reliability >0.5, based on convergent validity of Cronbach's alpha (CA) >0.7, composite reliability (CR)>0.708, and average variance of the extraction (AVE)>0.5. The second set is discriminant validity, AVE > exogenous item correlation values [70]. If all evaluation conditions are met, the Smart-PLS3 algorithm and iterations are alternatively performed. In this model, six iterations and procedures for assessment were included, and the outcome was that two indicators were deleted in each construct/group with a low factor load (<0.4) and that the average variance (AVE) errors extracted were improved to an acceptable level.

## Results

### Descriptive statistics of demographic sample characteristics

Descriptive analysis is critical in an all-inclusive view of data analysis. It allows researchers to understand the essential data dimensions, including the mean and standard deviation, as shown in Table 1. The participants' exposure was coded into male (*n = 184*) and female (*n = 125*) dichotomous responses of gender (M = 0.60; S. D = 0.482). Regarding entrepreneurial intention (M = 0.65; S. D = 0.477), prior farming exposure with (M = 0.63; S. D = 0.482) and prior entrepreneurial knowledge (M = 0.62; S. D = 0.485) revealed the highest mean and standard deviation, respectively. However, previous entrepreneurial exposure (M = 0.58; S. D = 0.494) and parental entrepreneurial orientation (M = 0.57; S. D = 0.496) revealed the lowest mean and standard deviation for the student's sample.

## Validity and reliability results

In the whole research sample, the overall construct EI, loaded with 4-items, exhibited composite reliability (CR = 0.87), average variance extraction (AVE = 0.63), and CA = 0.80. However, the attitude toward behavior was evaluated with three questions, yielding an AVE = 0.68 and composite reliability of (0.87) in the entire sample, including a Cronbach's alpha (CA) = 0.76. As a result, the subjective norm constructs were loaded with three items, resulting in CA = 0.61 and AVE = 0.54. According to [71], if the value of AVE is often too stringent, composite reliability (CR) alone may demonstrate the reliability of the model. As a result, the composite reliability of SN = 0.76 for the entire sample, and the perceived behavioral control constructs were loaded with three items, with construct reliability of 0.82. The AVE was 0.61, with a CA = 0.69 for the entire sample. As a result, the individual personality characteristics loaded with three items had a composite reliability of 0.83, an AVE = 0.63, and a CA = 0.70 in the whole sample. Similarly, entrepreneurial education constructs loaded with 3-items and evaluating the highest CR = 0.88, AVE = 0.71, and CA = 0.81 of the respondents' entrepreneurial intention are presented for the whole sample in Table 2.

The goodness-of-fit index test demonstrates that the equal constraint of specific structural parameters is continuously liberated, and the fitting effect of the overall model is retested [70]. Cutoff criteria for fit indices in covariance structure analysis: Conventional criteria vs. novel alternatives support chi-square (617.454), CFI (0.93), and SRMR cutoff measures (0.07).

## Discriminant validity criterion

Following each variable's validation and reliability testing, a PLS (SEM) was used to investigate the concurrent existence relations while considering variables affecting intention to the resultant of PLS (SEM). It was verified that the route coefficients for the set of criteria have

**Table 2. Constructs reliability and validity test measurement.**

| Constructs | Items | FL | CA | rho_A | (CR) | (AVE) |
|---|---|---|---|---|---|---|
| **Entrepreneurial Intention (EI)** | EI1 | 0.821 | 0.804 | 0.807 | 0.872 | 0.63 |
| | EI2 | 0.836 | | | | |
| | EI3 | 0.863 | | | | |
| | EI4 | 0.750 | | | | |
| **Attitude Toward Behavior (ATB)** | ATB1 | 0.718 | 0.761 | 0.782 | 0.865 | 0.683 |
| | ATB2 | 0.829 | | | | |
| | ATB3 | 0.920 | | | | |
| **Subjective Norm (SN)** | SN1 | 0.791 | 0.609 | 0.627 | 0.776 | 0.537 |
| | SN2 | 0.686 | | | | |
| | SN3 | 0.718 | | | | |
| **Perceived Behavioral Control (PBC)** | PBC1 | 0.739 | 0.689 | 0.703 | 0.824 | 0.610 |
| | PBC2 | 0.816 | | | | |
| | PBC3 | 0.786 | | | | |
| **Personality Traits (PT)** | PT1 | 0.828 | 0.702 | 0.704 | 0.834 | 0.627 |
| | PT2 | 0.797 | | | | |
| | PT3 | 0.749 | | | | |
| **Entrepreneurial Education (EE)** | EE1 | 0.821 | 0.813 | 0.923 | 0.878 | 0.706 |
| | EE2 | 0.836 | | | | |
| | EE3 | 0.863 | | | | |

Notes: FL = Factor Loadings; CA = Cronbach Alpha CR = Composite Reliability; AVE = Average Variance Extraction.

Table 3. Measurement of Squared Correlation and Discriminant Validity.

| Constructs | (ATB) | (EE) | (EI) | (PBC) | (PT) | (SN) |
|---|---|---|---|---|---|---|
| Attitude Toward Behavior (ATB) | *0.826* | | | | | |
| Entrepreneurial Education (EE) | 0.288 | *0.840* | | | | |
| Entrepreneurial Intention (EI) | 0.501 | 0.476 | *0.794* | | | |
| Perceived Behavioral Control (PBC) | 0.745 | 0.347 | 0.561 | *0.781* | | |
| Personality Traits (PT) | 0.555 | 0.287 | 0.696 | 0.697 | *0.792* | |
| Subjective Norm (SN) | 0.554 | 0.403 | 0.574 | 0.558 | 0.569 | *0.733* |

Diagonal values represent the square root of AVE.

discriminant validity. The Fornell-Lacker criterion assessment compares the square root of the AVE to exogenous construction associations [72]. The findings of the squared correlation criteria and discriminant validity are presented in Table 3.

Table 3 shows that exogenous factors such as ATB, PBC, SN, EE, and PT. The square root value of AVE is more important than other correlation values with external constructs and is believed to be adequate for establishing discriminant validity [73]. However, the square root values of the ATB and EE constructions were more significant than the PT, PBC, and EI values, with minor variations can be found in the extracted results. The overall discriminant validity of the model is well established, with substantial variations in the AVE's square root endured [70].

## Structural equation model measurement

PLS-SEM was used to verify the hypothetical model's accuracy and validate the structural equation measurement, and 2000 subsamples were bootstrapped. These connections were explained using bootstrap methods and t-statistics. The path coefficient and determinant coefficient ($R^2$) were discussed extensively. H1(a, b, c), H2(a, b, c), and H3(a, b, c) structural model path coefficients were assessed (see Fig 2 and Table 4). Tables 5–7 provide the mediation study of specific indirect and total indirect effects represented by bootstrapping techniques to testify the suggested H4 (a,b), H5 (a,b), and H6 (a, and b) of male and female samples, respectively.

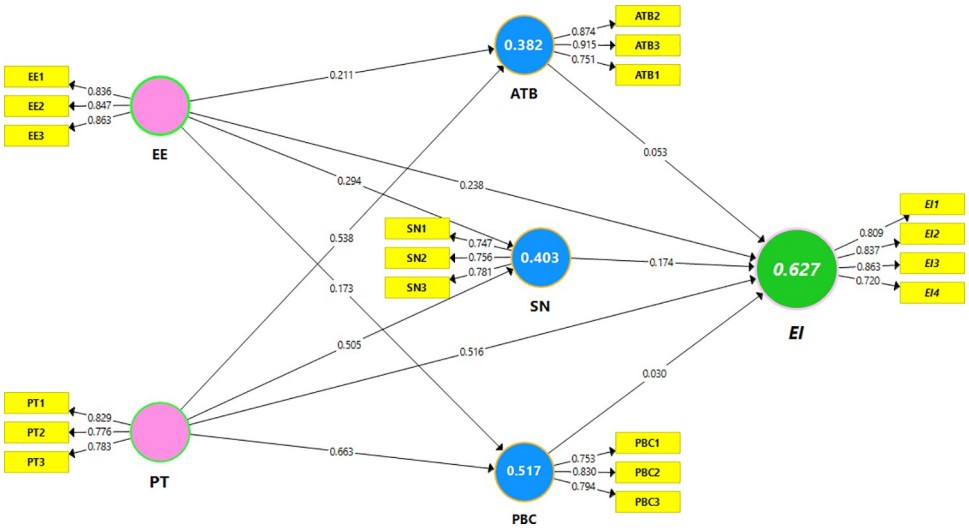

Fig 2. PLS-SEM Co-efficient path model estimation.

**Table 4. Estimates of Path Co-efficient in a Model.**

| Relationships | Male | Female | Full Sample | t- Statistics | p-Value | Hypotheses |
|---|---|---|---|---|---|---|
| **Entrepreneurial Intention** | | | | | | |
| **ATB→EI** | 0.054 | 0.145 | 0.083 | 0.991 | 0.322 | H1a |
| **SN→EI** | 0.171 | 0.135 | 0.146 | *2.181* | *0.029*\* | H1b |
| **PBC→EI** | 0.026 | -0.100 | -0.035 | 0.384 | 0.701 | H1c |
| **Entrepreneurial Education** | | | | | | |
| **EE→ATB** | 0.209 | 0.084 | 0.138 | *2.854* | 0.004\*\* | H2a |
| **EE→SN** | 0.277 | 0.120 | 0.235 | *3.652* | 0.000\*\*\* | H2b |
| **EE→PBC** | 0.171 | 0.230 | 0.159 | *3.796* | 0.000\*\*\* | H2c |
| **EE→EI** | 0.303 | 0.216 | 0.299 | *6.763* | 0.000\*\* | H2d |
| **Personality Traits** | | | | | | |
| **PT→ATB** | 0.540 | 0.494 | 0.516 | *11.011* | 0.000\*\*\* | H3a |
| **PT→SN** | 0.510 | 0.552 | 0.504 | *9.534* | 0.000\*\*\* | H3b |
| **PT→PBC** | 0.666 | 0.590 | 0.654 | *17.752* | 0.000\*\*\* | H3c |
| **PT→EI** | 0.656 | 0.584 | 0.612 | *17.508* | 0.000\*\*\* | H3d |

*Notes*: Sig

† $p < 0.100$

\* $p < 0.050$

\*\* $p < 0.010$

\*\*\* $p < 0.001$ and *n.s.*\* non-significant.

To investigate the hypothesis of H7 in Table 8, the authors utilized multi-group analysis (MGA) to examine the non-statistical hypothetical paths between the male and female models. Consequently, the findings showed that our study model parameters were equal among both groups, and no gender disparities were discovered. The results also showed that the variances were related to the observed heterogeneity. There may be non-observed heterogeneity; therefore, these heterogeneities are unlikely to influence any of the pre-specified variables [74].

## Hypothesis testing

A bootstrapping method was used to verify the proposed hypotheses and the significance of the projected correlation between the TPB antecedents and EE and PT on EIs (i.e., H1, H2, and H3), with 2,000 sub-sample relationship values and their significance level, considered in PLS-SEM analysis and the variance explanation [50]. Table 4, Figs 2 and 3 show the significant findings of each predicted connection.

The explained variance of the whole model, reflecting the EI $R^2$ = 59% for the total sample, ATB 33%, SN 39%, and PBC 51%, explained TPBs' three aspects of variation on the entrepreneurship intentions of respondents. The primary goal of statistical analysis was to differentiate between the two sub-samples. As a result, SN demonstrated a positive and substantial level in enlightening EI about H1, while social norm components support H1b\* only in the overall model; there is no significant difference between men and women.

The detection of H2 and EE showed a pledging effect of TPB dimensions; its impact on EI was favorable across all mediation factors. It has a good outcome, and its coefficient is both positive and significant for the agricultural student sample in terms of entrepreneurial intentions. Thus, H2a\*\*, H2b\*\*\*, H2c\*\*\*, and H2d\*\*\* are supported.

Comparably, the favorable effect of personality behaviors on the three TPB dimensions and EI findings indicated that H3a\*\*\*, H3b\*\*\*, H3c\*\*\*, and H3d\*\*\* are significant across all samples.

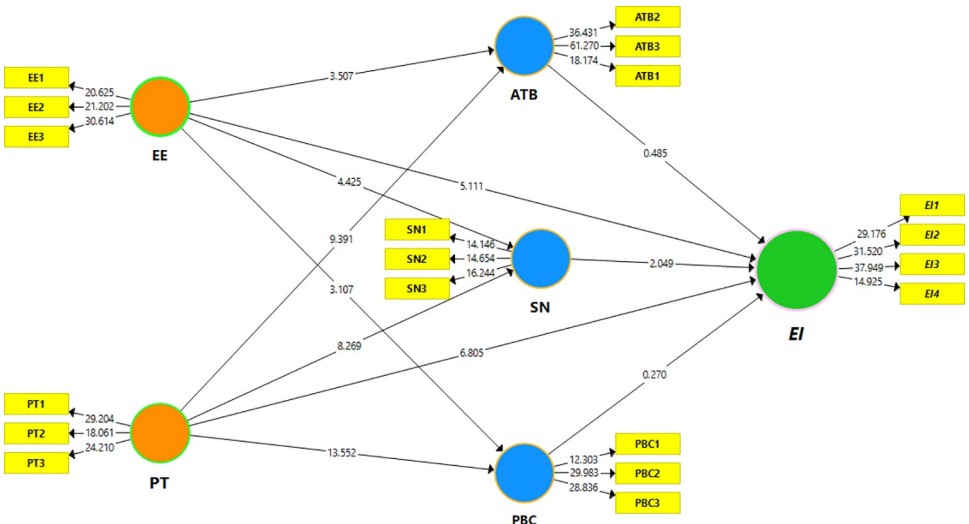

**Fig 3. PLS-SEM bootstrapping model significant estimation.**

Individual attitudes, perceived control behavior, and social norms played a crucial role in developing unique personalities and entrepreneurial indication, indicating that their relationship positively influenced and depicted a significant association; thus, all proposed hypotheses are supported. While gender was used as a control variable to determine the effect on EI in the agrarian sector, both gender groups were significantly impacted and linked in all samples, as shown in Tables 5–7.

## Analysis of meditation effects

Mediation impacts its three categories: indirect mediation, direct non-mediation only and, no non-mediation effect; complementary mediations and partial mediation were described as criteria utilized the method of [48] also regulated the mediation type as complete mediation and in partial mediation and complementary mediation only [50].

To assess the mediation paths function of the three antecedents of TPBs, namely ATB, SN, and PBC, considering extended factors as entrepreneurial education and personalities traits to predict the entrepreneurship intentions. The mediation analysis to test H5–H6 showed that subjective norms directly affect EIs were positively mediated in the paths of EE →SN →EI and

**Table 5. Structural specific indirect effects of the female sample.**

| Relationships | Coefficients (β) | Lower Threshold | Upper Threshold | t- Statistics | p-Value | Annotation |
|---|---|---|---|---|---|---|
| EE→ATB→EI | 0.012 | -0.012 | 0.063 | 0.618 | 0.536 | H4a |
| PT→ATB→EI | 0.072 | -0.027 | 0.182 | 1.323 | 0.186 | H4b |
| | | | | | | |
| EE→SN→EI | 0.016 | -0.026 | 0.062 | 0.757 | 0.449 | H5a |
| PT→SN→EI | 0.074 | -0.035 | 0.206 | 1.211 | 0.226 | H5b |
| | | | | | | |
| EE→PBC→EI | -0.023 | -0.092 | 0.044 | 0.681 | 0.496 | H6a |
| PT→PBC→EI | -0.059 | -0.208 | 0.125 | 0.698 | 0.485 | H6b |

Note: p- value is significant at the < 0.05. or

*p < 0.05

**p < 0.01and

***p = 0.001, Direct effects and Indirect effects: Bootstrapping: 2000 iterations and 0.95 bias-corrected.

**Table 6. Structural specific indirect effects of the male sample.**

| Relationships | Coefficients (β) | Lower Threshold | Upper Threshold | t- Statistics | p-Value | Annotation |
|---|---|---|---|---|---|---|
| EE→ATB→EI | 0.011 | -0.038 | 0.063 | 0.454 | 0.650 | H4a |
| PT→ATB→EI | 0.029 | -0.077 | 0.163 | 0.478 | 0.633 | H4b |
| | | | | | | |
| EE→SN→EI | 0.047 | 0.003 | 0.099 | 2.040 | *0.041** | H5a |
| PT→SN→EI | 0.087 | 0.006 | 0.192 | 1.812 | 0.070 | H5b |
| | | | | | | |
| EE→PBC→EI | 0.004 | -0.040 | 0.041 | 0.219 | 0.826 | H6a |
| PT→PBC→EI | 0.017 | -0.135 | 0.159 | 0.228 | 0.820 | H6b |

Note: p- value is significant at the < 0.05. or

*p < 0.05

**p < 0.01and

***p = 0.001, Direct effects and Indirect effects: Bootstrapping: 2000 iterations and 0.95 bias-corrected.

PT →SN →EI in the total sample, and the approaches of EE →SN →EI resulted in a significant and positive mediation role in the male model only. According to [50], bootstrapping with 2,000 sub-samples for the whole sample revealed the specific indirect and total outcome effects in Table 6. Hence, H5a* and H5b* are also supported in this study. To investigate each direct path's significance value and the principal function of an individually mediating variable in predicting the influence of EE and PT on EI, the variations and type of mediation can be perceived in the whole sample (see Tables 6 and 7). Both indirect and direct effects were substantial. However, they showed partial mediation results on EI. The paths direct and indirect effects are calculated to assess the nature of partial mediation followed by [75]; if the product's symbol is positive, a partial complementary intermediary is achieved [50].

Concerning H4, the TPB (ATB) dimension mediated the relationship between EE, PT, and EIs. For example, ATB mediated paths for EE-PT relationships on EIs nevertheless exhibited significant impacts on enterprise intention. In contrast, SN mediated and demonstrated a strong effect of EE and PT on EI in the male sample, directly or indirectly explaining EI via SN. Thus, H5a and H5b demonstrated significant results and support for the entire model. ATB mediated from PT to EI, although in both male and female samples, it is not to accept H4a and H4b in the complete and split data sets.

The PBC mediation paths function also impacted EE and PT with regard to adopting EIs in H6. The whole sample does not have a mediatory impact on EE and EI, as the male and female models do not support H6a and H6b shown in Table 7. and no mediation with a positive sign has been seen regarding the H6a connection between EE and EI, where PT and EIs discovered

**Table 7. Total indirect effects Male vs. Female.**

| | Female | | | Male | | |
|---|---|---|---|---|---|---|
| Relationships | Coefficients (β) | t- Statistics | p-Value | Coefficients (β) | t- Statistics | p-Value |
| EE→EI | 0.005 | 0.142 | 0.887 | 0.063 | *2.177* | 0.030** |
| PT→EI | 0.087 | 1.496 | 0.135 | 0.133 | *2.270* | 0.023** |

Note: p- value is significant at the < 0.05. or

*p < 0.05

**p < 0.01and

***p = 0.001, Indirect effects: Bootstrapping: 2000 iterations and 0.95 bias-corrected.

**Table 8. Multi-group parametric test difference Male vs. Female.**

| Constructs | Path Coefficient-diff | t-Value | p-Value |
|---|---|---|---|
| ATB -> EI | 0.084 | 0.531 | 0.710 |
| EE -> ATB | 0.128 | 1.239 | 0.117 |
| EE -> EI | 0.032 | 0.340 | 0.382 |
| EE -> PBC | 0.055 | 0.594 | 0.724 |
| EE -> SN | 0.149 | 1.294 | 0.104 |
| PBC -> EI | 0.110 | 0.649 | 0.258 |
| PT -> ATB | 0.045 | 0.455 | 0.332 |
| PT -> EI | 0.033 | 0.285 | 0.392 |
| PT -> PBC | 0.072 | 0.879 | 0.192 |
| PT -> SN | 0.035 | 0.357 | 0.638 |
| SN -> EI | 0.033 | 0.255 | 0.401 |

Note: p-values <0.05 or > 0.95 show significant differences across the two samples.

the interposition in the total indirect effects of non-supportive relationships to H6a, and H6b does not accept the whole sample in this study.

## Multi-group analysis

PLS-MGA is used in conjunction with Henseler's MGA and the permutation technique to determine the difference between groups. PLS-MGA includes the validation of dimension invariance across the two groups, using all non-parametric utilized variables. However, the PLS-MGA output showed significant variations in the impact of ATB, PBC, and SN on EI (H5) between males and females at significance levels of 0.05 and 0.01, respectively (Tables 8 and 9). Our study result confirms the existence of a substantial difference in the impact of PT and EE on EI between males and females.

According to [70, 72], the composite model invariance (MICOM) used in PLS-MGA has been assessed to validate and the findings of this research were established in both steps and showed measurement invariance [70]. The structural and PLS algorithms were similar for both samples, confirming the specified invariance. For compositional invariances, an empirical distribution permutation method with a sampling level of a minimum of 5,000 permutations after a permutation process (Cu) at 0.05 is used; the original score correlation c and the correlations obtained by that method can be used to determine compositional invariances when c exceeds 5% of Cu [70, 72]. Both Henseler's MGA and the permutation technique

**Table 9. Permutation test for measuring invariance.**

| | Compositional Invariance C = 1 | | | | | Equal Mean Assessment | | | | | Equal Variance Assessment | | | | |
|---|---|---|---|---|---|---|---|---|---|---|---|---|---|---|---|
| Items | (CI) | C = 1 | P-Mean | 5% c | p-Values | Partial (MI) Established | Mean Difference | 2.5% | 97.5% | p-Values | Variance Difference | 2.5% | 97.5% | p-Values | Full (MI) Established |
| ATB | Yes | 0.997 | 0.997 | 0.991 | 0.308 | No | 0.134 | -0.215 | 0.229 | 0.240 | 0.302 | -0.26 | 0.269 | 0.024 | No |
| EE | Yes | 0.994 | 0.994 | 0.978 | 0.300 | No | -0.566 | -0.223 | 0.23 | | 0.349 | -0.26 | 0.272 | 0.011 | No |
| EI | Yes | 0.998 | 0.999 | 0.997 | 0.144 | No | -0.358 | -0.223 | 0.231 | 0.000 | 0.365 | -0.33 | 0.329 | 0.031 | yes |
| PBC | Yes | 0.997 | 0.997 | 0.989 | 0.438 | Yes | 0.065 | -0.226 | 0.232 | 0.570 | 0.131 | -0.26 | 0.284 | 0.348 | yes |
| PT | Yes | 1.000 | 0.999 | 0.997 | 0.750 | Yes | -0.044 | -0.218 | 0.229 | 0.710 | 0.184 | -0.28 | 0.31 | 0.222 | yes |
| SN | Yes | 0.997 | 0.99 | 0.966 | 0.648 | Yes | 0.060 | -0.222 | 0.226 | 0.610 | 0.153 | -0.26 | 0.27 | 0.266 | yes |

Note: Configure Invariance = (CI) measure invariance = (MI), Correlation = (C).

verified the significance/non-significance of the results differences, which strengthened the conclusions of the study. The primary purpose of the multi-group analyses was to validate H7 on whether the connection of TPB dimensions with business intent would be highly inclusive of two examples: PT on EIs. MGA was performed to evaluate the difference between these two instances to determine whether the constructs were statistically significant. The uniqueness of this research is seen in Table 8; the paths across both groups are not found to be significantly different. Overall, statistically significant differences between males and females were observed in the multi-group results. Thus, H7 is supported in this study.

## Discussion

The primary aim of this study was to evaluate an inclusive context for TPB antecedents to predict EI and determine the degree to which individuals want to be self-efficient among Pakistani respondents. Research has focused on the critical role of the TPB's mediation of attitudinal beliefs, that is, ATB, PBC, and SN via education, and personal traits, in general, which were explored through gender differences in entrepreneurship [6, 66].

Our study results showed intriguing structural connections regarding the relative significance of ATB, PBC, and SN components in predicting intention components. The direct and mediation effects between the variables included in the validated structural model indicate that the TPB dimensions are related to the intended dimensions in a unique manner. The findings show that the impacts of SN, EE, and PT were the most significant predictors of intention in terms of total effects, followed by ATB and PBC, which had the most negligible impact on intention. These findings emphasize the critical nature of disentangling the components of attitudes, PBC, and SN. This enables us to investigate whether each element of these variables is associated with entrepreneurial ambition differently. The study's findings showed that the model predicted the effect on EIs in a manner that was consistent with prior research [6, 66]. Therefore, the TPB's predictive ability was acknowledged in a previous study [2, 36, 76]. In terms of the impact of SN dimensions on intention dimensions, the majority of prior research indicates that ATB and PBC are better predictors of intent [13, 77]. Conversely, in our analysis, ATB and PBC revealed weak predictions of male and female intentions.

However, the intensity of EIs varies by country; the TPB model's extrapolative power is higher in an emerging economy context, demonstrating stronger EIs, consistent with the [2, 78] discovered significant differences while determining the role of EIs between men and women in Pakistan. The mediation effects of TPB antecedents on gender showed that all except one of the intention variables were gender invariant. Males were more likely than females to be committed to new business ventures. In line with [78], this finding indicates that women are less likely than men to act on their entrepreneurial intentions.

The current study aimed to assess the effects of EE and PT on TPB domains and EIs, either directly or indirectly. The study highlighted the importance of personality and EE in determining creative career and business stimulation among young entrepreneurs [2, 79], verifying the reputation of individual character in entrepreneurial entrance and endurance [6, 80], and clarifying environmental factors and their significance [47]. In our study influence of personality traits mediated by TPB as antecedents was examined favorably significant [6, 81, 82] also discovered significant variations in the paths coefficients among students who indicated more futuristic entrepreneurial desire.

Entrepreneurial education had a more significant effect on TPB dimensions among participants solely concerning intentions. These results show that expanded TPBs have a substantial impact on business behavior. However, in developing nations [6, 80], they acknowledged that personality and enterprise training go beyond describing personality features in their job

choices and decision-making. The distinctive PT are based on the sample's environmental, familial, and socio-economic disparities under examination; the mediation findings directly or indirectly affect the intentions [50]. The results also indicate that a gender view of entrepreneurship may be derived from gender preconceptions, given that men and women use different sources of support [6, 61]. The mainstream media and educators may provide additional information regarding entrepreneurship to the gender-neutral features of women and men [61]. The training and education of companies should also be customized to address the requirements of entrepreneurs, men, and women to assist entrepreneurs.

## Conclusion and recommendations

The present study investigates EI using conceptual and intentional models from the TPB and adopts an SEM approach. This cross-sectional study examined the network of relations among gender differences in individual PT, EE, and within the TPB antecedents, specifically ATB, SN, and PBC, and male and female aspirants' entrepreneurial intentions in Pakistan. The hypothesized model was tested using a multi-group structural equation analysis. The dimensions of the TPB constructs were disentangled and treated as latent variables that were, directly and indirectly, inferred from multiple indicators.

### Theoretical implications

This study contributes to the entrepreneurial literature in many ways by investigating the relationships between behavioral beliefs, attitudes, and intentions within the TPB. To the best of our knowledge, this is one of the first efforts to untangle the dimensions of the constructs under study and investigate the relationships between these dimensions using both direct and indirect measures of attitudes in males and females in Pakistani respondents. The findings of this study fully support this inclusively literature-driven approach to the critical function of TPB domains (i.e., ATB, PBC, and SN) to mediate the PT and EE paths on entrepreneurial intention variables. These findings are essential for entrepreneurs and educators. Their belief in becoming entrepreneurs may be strengthened by specific and practical training and instruction.

This study emphasizes the dynamic role of the support of family and friends in determining the entrepreneurial ability of prospective agricultural entrepreneurs, showing that concentrating solely on unique features is not sufficient to better understand entrepreneurship. Given this, an entrepreneurial social culture, particularly in Pakistan, must be developed. Incubators must be built by the government, tax reliefs introduced, and laws drafted to enable local risk capital companies and investors to start a new business. The government must take the initiative in this regard.

Moreover, given the importance and lack of practical skills among farmers, it is recommended that teaching practices and training courses be reviewed. Interaction and cooperation among universities, management departments, and organizations involved in this field be strengthened while adding practical approaches and identifying appropriate goals. Pakistan must create a stable and productive institutional entrepreneurship environment. Therefore, revising current regulations and infrastructure is suggested to clarify how young entrepreneurs rapidly engage in self-employment enterprises.

### Practical implications

The study's findings offer a foundation for policies to enhance entrepreneurial activity in academic performance and enable students to set up enterprises in an atmosphere conducive to the future. Endorsing business enterprises should be encouraged to utilize favorable

government policies. A business development plan must be launched, including establishing a business plan for Pakistan, developing companies, and creating a large and conducive business ecosystem for emerging entrepreneurs in Pakistan. These initiatives should trickle down to the university level and motivate students to consider entrepreneurship as a viable economic tool. This study can help university decision-makers design a holistic strategy for developing students' interest in agriculture. Business managerial institutes can provide awareness, build capacity, and set up steering committees to assist students with practical business ideas. As a result, measures need to be taken to increase agricultural enterprises' internal and external motivations by encouraging development in agriculture, as well as making agriculture more competitive, profitable, and sustainable to ensure greater participation of young people, especially starting from their futuristic intentions.

Along with the significant gender difference in perceived attitudinal beliefs favoring men and the significant direct effect of subjective norms on nascent entrepreneurship found in our study, the aforementioned mediating effect of the TPB on gender suggests that educators and policymakers should create and support an environment that fosters entrepreneurial intention-action translation.

In a developing nation, this study highlights the need to translate work values into the entrepreneurship domain and improve attitudes toward entrepreneurship to foster emerging entrepreneurs. To this end, promoting sustainable business practices via the provision of business opportunities to prospective entrepreneurs may act as a double-edged sword, seeding sustainability-driven work values and facilitating the adoption of entrepreneurial intents and start-up execution. Thus, enterprise possibilities will bolster entrepreneurial aspirations and encourage sustainable entrepreneurship in the nation. As a result, the research recommends promoting entrepreneurial-driven work values and suggests that a greater emphasis on work values is placed on speeding people's attitudes toward entrepreneurship.

## Limitations

This study has some limitations. The first pertains to the cross-sectional design, which limits the study's contribution to the literature on entrepreneurial aspirations. Further interpreted into actions, such as the longitudinal approach required. The sample consisted of agricultural students to explore specific examples only. As a result, the results should be interpreted cautiously and verified and reproduced in a more significant study with a diverse sample of students from other fields and institutions of higher learning. Second, the study participants expressed concern about utilizing their ideas as skill indicators because they are prone to bias and inaccuracy, creating business is much difficult [83] and did not dive into actual entrepreneurial activity [3]. However, this may not be relevant when a subjective evaluation of abilities is required. In contrast, personal ability evaluation may be a step toward potential creative variations in entrepreneurial aspirations, enhancing the study while enhancing result validity [6].

## Future outlook research

Future research could provide a more in-depth assessment of entrepreneurial behavior. This would help shed light on family members' and others' roles in an individual's closed network. This is especially essential in Pakistan, where familial connections and emotional bonds between family members are strong. The factors that prevent entrepreneurial purpose from translating into entrepreneurial action requires further study for insights into the desired professional decision in other regions of Pakistan. All the studied factors should be considered in future studies. Based on the study results, it can be concluded that personality and educational value are crucial. The results can help us better understand the entrepreneurial intention

configuration process. Therefore, leveraging the sector's enormous potential to transform the economy, this study is a unique step forward in entrepreneurial intent. Future studies may be necessary to examine how students' self-efficacy develops as they complete university courses. Environmental variables that may enhance creativity and entrepreneurial self-efficacy across genders and disciplines can offer helpful information, especially for researchers working in higher education environments. Future research can gain insight into the factors contributing to positive or negative attitudes about a behavior, perceived social pressure to engage in an activity in recognizing that a 'one-size-fits-all' approach to curricula may not be appropriate and that gender sensitive programming, particularly in relation to various levels of entrepreneurial settings, in conjunction with the development of women's entrepreneurial aspirations.

## Supporting information

**S1 Data.**
(TXT)

**S1 Questionnaire.**
(DOCX)

## Acknowledgments

The authors would like to express heartfelt gratitude to fellow comrades for their involvement and help with the research. We are also appreciative to all of the reviewers who gave valuable input on the manuscript and contributed in finishing this.

## Author Contributions

**Conceptualization:** Ghulam Raza Sargani, Mudassir Hussain.

**Data curation:** Ghulam Raza Sargani, Abbas Ali Chandio, Asif Ali, Muhammad Rizwan, Najeeb Ahmed Kaleri.

**Formal analysis:** Ghulam Raza Sargani, Abbas Ali Chandio, Mudassir Hussain.

**Investigation:** Ghulam Raza Sargani, Yuansheng Jiang.

**Methodology:** Ghulam Raza Sargani, Abbas Ali Chandio, Asif Ali.

**Project administration:** Ghulam Raza Sargani, Yuansheng Jiang.

**Resources:** Yuansheng Jiang.

**Software:** Ghulam Raza Sargani, Najeeb Ahmed Kaleri.

**Supervision:** Yuansheng Jiang, Deyi Zhou.

**Validation:** Yuansheng Jiang, Deyi Zhou, Muhammad Rizwan.

**Visualization:** Deyi Zhou.

**Writing – original draft:** Ghulam Raza Sargani, Mudassir Hussain, Asif Ali, Najeeb Ahmed Kaleri.

**Writing – review & editing:** Ghulam Raza Sargani, Abbas Ali Chandio, Mudassir Hussain, Muhammad Rizwan, Najeeb Ahmed Kaleri.

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
