## [Decision Letter · Decision Letter 0]

13 Sep 2021

PONE-D-21-23172How do Gender disparities in entrepreneurial aspirations emerge in Pakistan? An Approach to Mediational and Multigroup AnalysisPLOS ONE

Dear Authors,

Thank you for submitting your manuscript to PLOS ONE. After careful consideration, we feel that it has merit but does not fully meet PLOS ONE’s publication criteria as it currently stands. Therefore, we invite you to submit a revised version of the manuscript that addresses the points raised during the review process.

ACADEMIC EDITOR: Please see comments below

We look forward to receiving your revised manuscript.

Kind regards,

Dejan Dragan, PhD

Academic Editor

PLOS ONE

Whilst you may use any professional scientific editing service of your choice, PLOS has partnered with both American Journal Experts (AJE) and Editage to provide discounted services to PLOS authors. Both organizations have experience helping authors meet PLOS guidelines and can provide language editing, translation, manuscript formatting, and figure formatting to ensure your manuscript meets our submission guidelines. To take advantage of our partnership with AJE, visit the AJE website (http://aje.com/go/plos) for a 15% discount off AJE services. To take advantage of our partnership with Editage, visit the Editage website (www.editage.com) and enter referral code PLOSEDIT for a 15% discount off Editage services.  If the PLOS editorial team finds any language issues in text that either AJE or Editage has edited, the service provider will re-edit the text for free.

A clean copy of the edited manuscript (uploaded as the new *manuscript* file

 “No fund provision to this study”

Additional Editor Comments (if provided):

The reviewers have completed their review. They have adopted quite diverse decisions, from rejecting the paper all over to demand for minor revision. Accordingly, my desicion is: Major revision. AE DD

Reviewers' comments:

Reviewer's Responses to Questions

**Comments to the Author**

1. Is the manuscript technically sound, and do the data support the conclusions?

Reviewer #1: Yes

Reviewer #2: Yes

Reviewer #3: Partly

Reviewer #4: No

2. Has the statistical analysis been performed appropriately and rigorously? 

Reviewer #1: Yes

Reviewer #2: Yes

Reviewer #3: Yes

Reviewer #4: No

3. Have the authors made all data underlying the findings in their manuscript fully available?

Reviewer #1: Yes

Reviewer #2: Yes

Reviewer #3: Yes

Reviewer #4: Yes

4. Is the manuscript presented in an intelligible fashion and written in standard English?

Reviewer #1: Yes

Reviewer #2: Yes

Reviewer #3: Yes

Reviewer #4: No

5. Review Comments to the Author

Reviewer #1: I have completed my review of the Manuscript Number: PONE-D-21-23172.

Title: How do Gender disparities in entrepreneurial aspirations emerge in Pakistan? An Approach to Mediational and Multigroup Analysis

Overall: Although the concept sounds interesting, the research is done very well, and the paper is thoroughly examined accordingly to readers perceptive.

The title seems to be very innovative exciting. Still, I must have some questions regarding your Manuscript the Author(-s) present an empirical study exploring the direct influence of Personality Traits and Entrepreneurial Education on Entrepreneurial Intention, estimated the mediating role of attitudes toward behavior, subjective norm and perceived control behavior on Entrepreneurial Intention why did you choose indirect effects?

Form and Style and Grammar:

Across the literature review and other areas, the use of past tense is the norm or refers to research already accomplished?

Introduction: it is well articulated and evidently well-defined the structure and research questions

An abstract is well written and complete description of the study and the participants.

The literature review was oudated to read. It reads as a regurgitation of facts from many other articles. You are reviewing the literature, but it is in your thoughts and research supporting the literature. Although it flow and transition in many areas and effectively create a case for your study however, re-correct and revise.

Methods:

Author(-s) should clarify their procedure iso making the reader speculate on what has been done. How instruments and the support are good, but there is little detail on how you used them and why they were used. Data collection processed not much evident, how many samples you choose and what sampling strategy you applied can you explain with evidence? Why did you choose Smart PLS rather than AMOS?

Results:

However, although the results and tabulations are well managed and understandable to readers' apprehensions and viewpoints, it must be redesigned according to the format of the journal?

Discussion: whereas the discussion section can be extending more with UpToDate reviews and researches? Add some limitations implications and future prospects of the study. Hence, it is not possible to evaluate the contribution to knowledge development.

References:

In-text citations are not appropriately formatted and are used without first listing all the authors. Reference pages are not correctly formatted according to journals format?

Reviewer #2: • When I was given the opportunity to review the manuscript entitled: How do Gender Disparities in Entrepreneurial Aspirations Emerge in Pakistan: An Approach to Mediational and Multigroup Analysis.

• In addition to being very original and intriguing, the title is extremely catchy. As a consequence, I have a few concerns regarding the text as a result of this. Based on their findings, the authors provide empirical research in which they investigate the direct impact of personality characteristics on entrepreneurial intention and the indirect influence of subjective norm and perceive control behavior on entrepreneurial intention.

• The introduction is straightforward and concise in terms of form, style, and language, and the framework and study subjects are well identified. When writing an abstract, make sure it's well-written and includes a thorough explanation of the research and participants.

• Is the past tense usage ubiquitous in the literature review and other areas, or does it relate to previously completed research? The literature evaluation was made available for reading. It reads like a rehash of many other publications' information. While you are reading the material, your insights and research are substantiating the literature. Although it flows and transitions well in many sections and successfully builds a case for your research, it should be revised and re-corrected.

• Methods: Why did you select Smart PLS over AMOS as your method of choice? Instead of leaving the reader shady regarding how something was accomplished, the author(s) should describe their strategies and approach. Although the instruments and support are of high quality, little information is available on how and why they were used. Data collection methods are not completely clear; how many samples you selected and the selection methodology you employed are not entirely visible; are you able to provide evidence to support your decisions?

• However, even though the findings and tabulations are well-organized and easily understood in light of readers' concerns and perspectives, they must be modified to conform to the journal's format?

• Whereas the discussion area might be expanded with more up-to-date evaluations and researches? Include some of the study's shortcomings, consequences, and future opportunities. As a result, the contribution to knowledge growth cannot be quantified.

• Citations in the text: In-text citations are not properly structured, and they are utilized without first providing the names of all authors. Are reference pages not structured properly following the journal's format?

• Overall, I was ecstatic that while the idea is intriguing, the study is done very effectively. The article is quite extensively analyzed, following the readers' perceptions of the subject matter.

• Finally, keep in mind that I highly endorsed this manuscript; nevertheless, the authors can explain themselves above any issues; thus, I must assign this paper with minor revisions to get acceptance from the editor of the reputed journal.

Reviewer #3: acThe study aims to find out the “How do Gender disparities in entrepreneurial aspirations emerge in Pakistan? An Approach to Mediational and Multigroup Analysis”. The topic is interested in the field. However, some considerations need to be explained for improving the quality of the manuscript and after minor explanations and corrections, the paper could be published.

Reviewer #4: Though I am a non-native, still it was difficult for me to understand the communication in many places in this manuscript.

Abstract

1. Abstract is too long. It should be precise with the core information from the paper.

Introduction

I missed a solid problem motivation in introduction. Statements seemed to be not well integrated. Thus, it becomes difficult to grasp the knowledge-gap/problem which is addressed by this study.

1. Sounds strange: personality characteristics, innovating innovative approaches….

2. What is the elaboration of TPB (Page 4, line 100)?

Literature review

1. This section hardly reflected the theoretical background on the subject matter.

2. It appears that some components of the methodology are incorporated in this section.

Materials and method

1. I am confused about the sample size under the sub-section ‘Data collection procedure and participants.’

2. It is not mentioned how the samples were collected.

3. What are the core variables used in this study? How were those measured?

Results and Discussion

1. Results from a number of reliability and validity tests are incorporated (Table 2). I wonder if data were checked for suitability of running SEM.

2. I could hardly find the gender aspect in the bunch of analyses undertaken in this section.

3. It appears the findings are not well-articulated with discussion

6. PLOS authors have the option to publish the peer review history of their article (what does this mean?). If published, this will include your full peer review and any attached files.

Reviewer #1: No

Reviewer #2: No

Reviewer #3: No

Reviewer #4: No

---

## [Author Response · Author response to Decision Letter 0]

8 Oct 2021

Dear Reviewer,

Thank you very much for your consideration of our PONE-S-21-16449 manuscript. 

We have fully addressed each point and carefully described our response after each comment and questions elevated by respected reviewers:

We are going to respond to the editorial desk, and we anticipate that the modifications are adequate. If there are somewhat changes, please indicate new modifications that would revise and improve the manuscript and make it suitable for its acceptance.

In the revised manuscript version, the new sentences and paragraphs are in the main revised manuscript, With Track Changes (marked-up copy) and Without Track, Changes (clean copy) files to easily be recognized by the editorial desk. Whereas the response to reviewers are made in Red (questions raised by reviewers) and in Blue (Response to reviewers) are also attached here with.

Responses to Reviewers 

IReviewer #1: 

I have completed my review of the Manuscript Number: PONE-D-21-23172.

Title: How do Gender disparities in entrepreneurial aspirations emerge in Pakistan? An Approach to Mediational and Multigroup Analysis

Overall: Although the concept sounds interesting, the research is done very well, and the paper is thoroughly examined accordingly to readers perceptive.

The title seems to be very innovative exciting. Still, I must have some questions regarding your Manuscript the Author(-s) present an empirical study exploring the direct influence of Personality Traits and Entrepreneurial Education on Entrepreneurial Intention, estimated the mediating role of attitudes toward behavior, subjective norm and perceived control behavior on Entrepreneurial Intention why did you choose indirect effects?

Dear reviewer, thank you very much for your remarks regarding this study Researchers prefer structural equation modeling (SEM) over traditional analyses, which overlook the connections between latent components that are implicitly assessed through numerous measurement items and pathways (Bollen, 2014; Chin, 1998a). Among the two primary SEM techniques in use today, covariance-based and partial least squares (PLS), the latter is more appropriate than other analytical methods for a variety of reasons. To begin, the PLS-SEM approach simplifies the modeling of formative and reflective structures by allowing for the handling of second-order constructs (Chin, 1995, 1998b) (Wetzels et al., 2009). Second, it analyzes both the measurement and structural models concurrently (Wixom & Watson, 2001). Third, the PLS-SEM technique is often suggested when the multivariate normality criterion is violated in a dataset (Kock, 2020c).

While PLS-SEM is often used, it frequently produces results that enable multivariate normality, multicollinearity, common-method bias, and prediction validity testing.

Form and Style and Grammar:

Across the literature review and other areas, the use of past tense is the norm or refers to research already accomplished?

Dear reviewer, thank you very much for your comment. We added new citations the following statements and up to date relevant literature and appropriate range of cited sources in the literature review section have been incorporated in the main text.

Introduction: it is well articulated and evidently well-defined the structure and research questions

Dear reviewer, thank you very much for your comment regarding introduction section has been revised, the introductory part has also been re-structure and updated. It may explain the problem and issue of this analysis to provide the research primary and objectives; fresh modifications are produced to the text.

An abstract is well written and complete description of the study and the participants.

Thank you very much for your comments in the sense of abstract of the paper has been summarized, and succinct results have been updated and included into the final version of the manuscript as well.

The literature review was oudated to read. It reads as a regurgitation of facts from many other articles. You are reviewing the literature, but it is in your thoughts and research supporting the literature. Although it flows and transition in many areas and effectively create a case for your study however, re-correct and revise.

We appreciate your time in leaving a remark. The literature review statements and section have been updated with new citations, and the most current relevant material, as well as a sufficient range of cited sources from the literature review section, have been included into the main text.

Methods:

Author(-s) should clarify their procedure iso making the reader speculate on what has been done. How instruments and the support are good, but there is little detail on how you used them and why they were used. Data collection processed not much evident, how many samples you choose and what sampling strategy you applied can you explain with evidence? Why did you choose Smart PLS rather than AMOS?

We surveyed 400 graduate students from four of Pakistan's major institutions. Our choice of a population-restricted agribusiness students is justified by Pakistan's pronounced gender disparities in entrepreneurship. Our selection of simple random sampling strategy based on population restricted agribusiness students is supported by the Pakistani contexts distinctive gender inequalities in entrepreneurship. A total of 400 questions were posed, and 350 investigations were conducted, yielding a total of 309 relevant questions and a response rate of 77.25 percent, with 184 males (60%) and 125 females participating (40 percent) 

PLS-SEM is a technique for exploratory data analysis that utilizes primary or secondary data (Not suitable for CB-SEM-Hair 2015). The PLS method is appropriate for researchers with a prediction-oriented goal since it does not need normal data distribution and allows for small sample sizes (Chin & Newsted, 1999). Wherever CB-SEM is used, a larger sample size (minimum >400) is required. One of the primary benefits of PLS-SEM over CB-SEM is its ability to handle many independent variables concurrently, even when they exhibit multicollinearity (Hair, Ringle, & Sarstedt, 2011). Additionally, PLS-SEM gives R2 values and shows the importance of connections between constructs to illustrate the model's performance. On the other hand, CB-SEM is limited to path modeling (coefficient and CR). Bear in mind that PLS is more appropriate for prediction-based research, while CB-SEM is not appropriate for model fit therefore we choose SMART PLS-SEM approach to analysis our data.

Results:

However, although the results and tabulations are well managed and understandable to readers' apprehensions and viewpoints, it must be redesigned according to the format of the journal?

All of the tables and figures have been included in the main manuscript and their numerical and numbering have been updated. Table 1 to Table 9 and Figure 1 on page 9 to Figure 3 on page 18 have been included in the main text as well as their numerical and numbering have been changed.

Discussion: whereas the discussion section can be extending more with UpToDate reviews and researches? Add some limitations implications and future prospects of the study. Hence, it is not possible

All of the results in the discussion section and contributions from the research have been included in the discussion section, which allows us to compare our findings with those of other recent studies. Our findings provide answers to the study's pertinent issues and achieve the study's primary goals. e to evaluate the contribution to knowledge development. Also, limitations theoretical and empirical implications and future prospects of the study have been added in the conclusion section 

References:

In-text citations are not appropriately formatted and are used without first listing all the authors. Reference pages are not correctly formatted according to journals format?

Throughout references are used to refer to all correctly formatted references without first identifying all of the authors. Citation pages are revised and restructured to conform to the journal's style.

 

Reviewer #2:

When I was given the opportunity to review the manuscript entitled: How do Gender Disparities in Entrepreneurial Aspirations Emerge in Pakistan: An Approach to Mediational and Multigroup Analysis.

• In addition to being very original and intriguing, the title is extremely catchy. As a consequence, I have a few concerns regarding the text as a result of this. Based on their findings, the authors provide empirical research in which they investigate the direct impact of personality characteristics on entrepreneurial intention and the indirect influence of subjective norm and perceive control behavior on entrepreneurial intention.

• The introduction is straightforward and concise in terms of form, style, and language, and the framework and study subjects are well identified. When writing an abstract, make sure it's well-written and includes a thorough explanation of the research and participants.

Dear Reviewer. Thank you for your feedback on the introduction section, which has been changed. The introductory portion has also been re-structured and updated as a result of your feedback. As a result of this analysis, it is possible to describe the problem and issue in order to give the research primary and goals; new changes are made to the text. We have addressed each issue and provided a thorough explanation of our answer following each remark. The new phrases are highlighted in yellow in the updated version of the document, making it easy for the reviewers to see where they have been added.

• Is the past tense usage ubiquitous in the literature review and other areas, or does it relate to previously completed research? The literature evaluation was made available for reading. It reads like a rehash of many other publications' information. While you are reading the material, your insights and research are substantiating the literature. Although it flows and transitions well in many sections and successfully builds a case for your research, it should be revised and re-corrected.

In the literature review section, the most current relevant material has been included, as has an adequate range of referenced references from a variety of sources. When it comes to transitioning into self-employment, the literature review section looks at the variables that affect it, with special emphasis given to the variations between males and females.

• Methods: Why did you select Smart PLS over AMOS as your method of choice? Instead of leaving the reader shady regarding how something was accomplished, the author(s) should describe their strategies and approach. Although the instruments and support are of high quality, little information is available on how and why they were used. Data collection methods are not completely clear; how many samples you selected and the selection methodology you employed are not entirely visible; are you able to provide evidence to support your decisions?

PLS-SEM is an exploratory data analysis method that makes use of either primary or secondary data sources (Not suitable for CB-SEM-Hair 2015). Because it does not need normal data distribution and allows for small sample sizes, the PLS technique is well suited for researchers with a prediction-oriented objective (Chin & Newsted, 1999). CB-SEM is only applicable in situations in which a higher sample size (minimum >400) is needed. One of the most significant advantages of PLS-SEM over CB-SEM is its capacity to handle a large number of independent variables at the same time, even when they show multicollinearity (Hair, Ringle, & Sarstedt, 2011). To further highlight the model's effectiveness, PLS-SEM provides R2 values and demonstrates the significance of linkages between constructs, among other things. CB-SEM, on the other hand, is restricted to path modeling alone (coefficient and CR). Because PLS is more suitable for prediction-based research, but CB-SEM is not appropriate for model fit, we have chosen a SMART PLS-SEM method for our data analysis.

Our poll included almost 400 graduate students from four major Pakistani universities. The fact that there are significant gender inequalities in entrepreneurship in Pakistan lends credence to our selection of agribusiness students from a limited sample of the country's population. The particular gender disparities in entrepreneurship that exist in the Pakistani context justify the use of a basic random sample method in this research, which is based on agribusiness students from a restricted population. 400 questions were asked, and 350 investigations were carried out, yielding a total of 309 pertinent questions and a response rate of 77.25 percent, with 184 males (60 percent) and 125 women participating in the research (40 percent).

• However, even though the findings and tabulations are well-organized and easily understood in light of readers' concerns and perspectives, they must be modified to conform to the journal's format?

All tables and figures have been included into the main text, and their numerical and numbering conventions have been changed. Tables 1 to 9 and Figures 1 to 3 on pages 9 to 18 have been included into the main text, and their numerical and numbering conventions have been modified accordingly.

• Whereas the discussion area might be expanded with more up-to-date evaluations and researches? Include some of the study's shortcomings, consequences, and future opportunities. As a result, the contribution to knowledge growth cannot be quantified.

The comparison of our findings to those of other current studies due to the fact that all of the results and all of the study's contributions were included in the discussion section. It is our results that have provided answers to relevant research issues and have enabled us to accomplish the main objectives of the study, which were to evaluate the study's contribution to knowledge creation and dissemination. Also included are discussions of the study's shortcomings, theoretical and empirical consequences, and future possibilities in the conclusion part of the paper.

• Citations in the text: In-text citations are not properly structured, and they are utilized without first providing the names of all authors. Are reference pages not structured properly following the journal's format?

Throughout the manuscript, the bibliography and document citation have been restructured and are utilized without first mentioning that all authors current and up to date citations are added. The reference pages have been re-formatted to be more readable. It has been utilized throughout the remainder of the citations and in the final list of references. The paper has been thoroughly examined, and we have found no additional mistakes of this kind in it.

• Overall, I was ecstatic that while the idea is intriguing, the study is done very effectively. The article is quite extensively analyzed, following the readers' perceptions of the subject matter.

• Finally, keep in mind that I highly endorsed this manuscript; nevertheless, the authors can explain themselves above any issues; thus, I must assign this paper with minor revisions to get acceptance from the editor of the reputed journal.

With all due respect and attention for your remark, we have examined the section and feel that we have identified the major limits of our study, to which we have included recommendations for future research. We thank you for your time and consideration. We have addressed each issue and provided a thorough explanation of our answer following each remark. The new phrases are highlighted in yellow in the updated version of the document, making it easy for the reviewers to see where they have been added.

 

Reviewer #3: 

The study aims to find out the “How do Gender disparities in entrepreneurial aspirations emerge in Pakistan? An Approach to Mediational and Multigroup Analysis”. The topic is interested in the field. However, some considerations need to be explained for improving the quality of the manuscript and after minor explanations and corrections, the paper could be published.

Dear reviewer, thank you very much for your suggestion. We added that sections and have re-organization and revised of the abstract, introduction, literature review study framework, hypothesis development, material and methods, results and analysis discussion and contribution and conclusion sections have also been completed. 

 

Reviewer #4: 

Though I am a non-native, still it was difficult for me to understand the communication in many places in this manuscript.

Abstract

1. Abstract is too long. It should be precise with the core information from the paper.

The abstract of the paper has been summarized, and succinct results have been updated and integrated into the text as well as the body of the manuscript.

Introduction

I missed a solid problem motivation in introduction. Statements seemed to be not well integrated. Thus, it becomes difficult to grasp the knowledge-gap/problem which is addressed by this study.

A re-organization and revision of the introductory section has also been completed. In order to provide the study's goals and objectives, it may explain the problem and issue of this analysis; new modifications are produced to the text as a result to grasp the knowledge-gap problem which is addressed by this study.

1. Sounds strange: personality characteristics, innovating innovative approaches….

2. What is the elaboration of TPB (Page 4, line 100)?

Personality characteristics are almost as effective as the more numerous lower-level, specialized qualities in predicting and explaining real behavior. Understanding a key employee is just as critical as understanding its operations and procedures. Understanding the personality components that influence subordinates' behavior is critical information for management because it can be used to determine the type of assignments that should be given, the manner in which motivation should be pursued, the team dynamics that may arise, and the best way to approach conflict and/or praise when applicable. However whole paragraph has revised accordingly.

Where as the TPB stands for the Theory of Planned Behavior was first created by Ajzen in 1991 to represent the connection between an individual’s attitude toward a specific action and the actual behavior.

Literature review

1. This section hardly reflected the theoretical background on the subject matter.

2. It appears that some components of the methodology are incorporated in this section.

The most recent relevant literature has been incorporated, as well as an appropriate range of cited sources in the literature review section. The literature review section examines the factors that influence transitions into self-employment, with particular attention paid to differences between men and women.

Materials and method

1. I am confused about the sample size under the sub-section ‘Data collection procedure and participants.’

2. It is not mentioned how the samples were collected.

3. What are the core variables used in this study? How were those measured?

We polled 400 graduate students from four major Pakistani universities. The fact that Pakistan has significant gender inequalities in entrepreneurship justifies our selection of a population-restricted agribusiness students. The Pakistani context's unique gender disparities in entrepreneurship justify our use of a simple random sample method based on population restricted agribusiness students. A total of 400 questions were asked, and 350 investigations were carried out, giving a total of 309 pertinent questions and a response rate of 77.25 percent, with 184 men (60 percent) and 125 females participating (40 percent).

This study's questionnaire was based on a 5-point Likert scale and modified from previous research on theory of planned behavior (TPBs). It goes from 1 (strongly disagree) to 5 (strongly agree) (strongly agree). The constructs Entrepreneurial Intention (EI) and Perceived Behavioral Control (PBC) were evaluated using five items from (Liñán & Chen, 2009);, while attitudes toward behavior (ATB) were examined using five questions from (Kolvereid & Isaksen, 2006). [22]. Finally, five items from (Hao Zhao et al., 2010) were used to assess Entrepreneurial Education (EE) and five questions to assess Personality Traits (PT). 

Whereas the control variables Entrepreneurial Intention (EIs) has been linked to various control factors (Liñán & Chen, 2009);. To determine gender, the author used dummy-coded variables for Entrepreneurial Intention (EI), Prior Entrepreneurial Knowledge (PEK), Parental Entrepreneurial Exposure (PEX), Prior Entrepreneurial Exposure (PEE), and Prior Farming Exposure (0 = female; 1 = male) and (1=Yes;0=No) for Entrepreneurial Intention (EI), Prior Entrepreneurial Knowledge (PEK), Parental Entrepreneurial Exposure (PEX), Prior Entrepreneurial Exposure (PFE). Whereas the gender disparities in entrepreneurial activity have been confirmed by the study of (Arshad et al., 2016).

Results and Discussion

1. Results from a number of reliability and validity tests are incorporated (Table 2). I wonder if data were checked for suitability of running SEM.

2. I could hardly find the gender aspect in the bunch of analyses undertaken in this section.

3. It appears the findings are not well-articulated with discussion

In order to accomplish this research objective, the author used Smart PLS 3. (Henseler et al., 2016) The structural equation model (SEM) is a precise technique that may be used to assess social, behavioral, and agricultural sciences in order to experimentally verify the relationship between model variables. There are many ongoing and preliminary stages involved in the adoption of this model. These phases include a description of the theoretical model to be tested, parameter estimation, and evaluation. It was decided to assess the reflective measurement model based on two sets of criteria. In order to support the validity of the questionnaire, measuring methods that have previously been tested and validated should be employed (Wang et al., 2015). Additionally, the author conducted preliminary survey testing on 30 respondents to ensure that the questions in this research were both understandable and trustworthy prior to distributing the survey. It has shown that certain problems have been moved and changed in order to simplify the study.

Our findings compared to those of other recent research is feasible since we included all of the results and all of the study's contributions in the discussion section. Our findings provide answers to pertinent research questions and allow us to achieve the study's primary goals, namely evaluating the study's contribution to knowledge production and dissemination. Additionally, the conclusion section discusses the study's limitations, theoretical and empirical implications, and future prospects.

We hope that the changes that have been made are satisfactory. If not, please specify any further changes that would be necessary to enhance the text and make it acceptable for submission.

Thanking in anticipation

---

## [Editor Report · Decision Letter 1]

10 Nov 2021

How do gender disparities in entrepreneurial aspirations emerge in Pakistan? an approach to mediation and multi-group analysis

PONE-D-21-23172R1

Dear Authors,

We’re pleased to inform you that your manuscript has been judged scientifically suitable for publication and will be formally accepted for publication once it meets all outstanding technical requirements.

Kind regards,

Dejan Dragan, PhD

Academic Editor

PLOS ONE

Additional Editor Comments (optional):

The authors have very carefully and adequately corrected their manuscript due to the instructions of reviewers. Accordingly, the AE's recommendation is: An acceptance of the paper.
---

## [Editor Report · Acceptance letter]

19 Nov 2021

PONE-D-21-23172R1 

How do gender disparities in entrepreneurial aspirations emerge in Pakistan? an approach to mediation and multi-group analysis 

Dear Dr. Sargani:

I'm pleased to inform you that your manuscript has been deemed suitable for publication in PLOS ONE. Congratulations! Your manuscript is now with our production department. 

Kind regards, 

on behalf of

Dr. Dejan Dragan 

Academic Editor

PLOS ONE